# Gene-centric evaluation of causal variant prediction for DNA models

Chantriolnt-Andreas Kapourani [1]   Alice Del Vecchio [1]   Agnieszka Dobrowolska [1]   Andrew Anighoro [1]
Edith M. Hessel [1]   Lindsay Edwards [1]   Cristian Regep [1]

## Abstract

DNA models hold significant potential for linking genetic variation to transcriptional regulation, which is crucial for understanding disease mechanisms at the genetic and molecular level and developing targeted therapies. Supervised approaches, such as Enformer and Basenji, have shown promising results in predicting causal variants. Recently, self-supervised models like Nucleotide Transformer and HyenaDNA have made remarkable advancements, with variant-centric benchmarks suggesting competitive performance on the variant effect prediction task. In this study, we propose to evaluate models also on gene-centric benchmarks, which often are of higher relevance to the genetics community for mapping causal variants to affected genes.

## 1. Introduction

The advent of high-throughput sequencing technologies has granted us unprecedented access to the entire human genome, revolutionizing our understanding of genetics and molecular biology. Biobanks, such as the UK Biobank (UKBB), offer vast repositories of individual genomes (Halldorsson et al., 2022). These resources provide extensive training data, but also play a crucial role in identifying genetic variations associated with traits and diseases. Most genetic variations linked to diseases occur outside of genes, in the non-coding genome (Uffelmann et al., 2021). The challenge lies in linking non-coding variations to downstream effects on specific genes, which leads to a better understanding of diseases, and in turn to drug targets. Notably, drugs developed for targets with genetic links have been shown to have more than twice the success rate in clinical trials (Minikel et al., 2024).

Traditionally, supervised models using convolutions and/or

---

[1]Relation, Regent's Place, 338 Euston Road, London NW1 3BG. Correspondence to: Cristian Regep <cristian@relationrx.com>.

*Accepted at the 1st Machine Learning for Life and Material Sciences Workshop at ICML 2024*. Copyright 2024 by the author(s).

transformer layers have been the standard approach for modeling DNA. These models were trained to predict epigenetic and transcriptional profiles directly from DNA sequences. Examples include ExPecto (Zhou et al., 2018), Enformer (Kelley, 2020), and Basenji (Avsec et al., 2021). These models have progressively improved the modeling of genomic data and the prediction of genetic variation effects on gene expression by leveraging ever-increasing receptive fields (Linder et al., 2023). Additionally, these models have demonstrated the ability to predict causal variants (Avsec et al., 2021).

Concurrently, the broader field of NLP research has focused on developing approaches for training transformer architectures using masked and causal language models (Zhao et al., 2023). In the past year, these techniques have also been applied to model genetic data, ushering in a new era of genomic language models (gLMs). Recent models, such as the Nucleotide Transformer (Dalla-Torre et al., 2023), HyenaDNA (Nguyen et al., 2024b), Mamba (Gu & Dao, 2023), and Evo (Nguyen et al., 2024a), are trained using masked or causal language modeling with context lengths ranging from 6,000 bp to 1 Mbp, utilizing up to 7 billion parameters. This marks a distinctive and significant leap in the scale of DNA models.

The benchmarks for causal variant effect prediction in these models has predominantly emphasized variant classification rather than the association of these variants with downstream affected genes. This methodology aligns with the approaches used in Avsec et al. (2021) and Kelley (2020). Establishing the link between variants and their respective affected genes presents significant challenges. One major complexity arises from Genome-Wide Association Studies (GWAS), which often report numerous non-causal single nucleotide polymorphisms (SNPs) due to linkage disequilibrium, resulting in many false positives (Uffelmann et al., 2021). Additionally, tissue expression data is often collected separately from genetic data, necessitating the use of external datasets, such as those from GTEx (Consortium, 2020), for accurate association between a variant and its transcriptional effect. Developing ab initio methods, or techniques that function with minimal data, would be highly beneficial in this context.

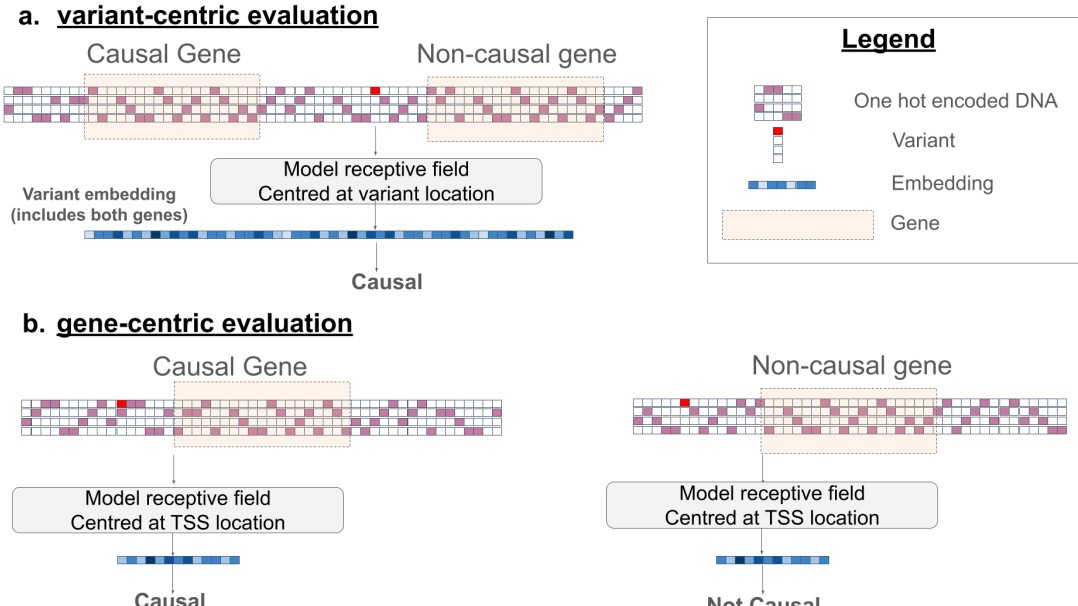

*Figure 1.* **a.** The challenge with variant-centric evaluation is that using the entire sequence embedding does not provide the ability to differentiate between specific genes. **b.** In contrast, gene-centric evaluation utilizes only the embedding of the TSS, enabling the prediction of individual downstream genes affected by the variant.

Despite these challenges, advancements in genomic language models hold considerable promise for enhancing our ability to link genetic variations to their functional consequences. The integration of these models with biobank data could lead to significant breakthroughs in understanding the genetic basis of disease and the development of more effective therapies.

## 2. Gene-centric causal variant prediction

Kao et al. (2024) released a benchmark study of current state-of-the-art models using an established ground-truth dataset (see Section 4.1). Their findings suggest that gLMs exhibit similar performance on causal variant prediction as supervised models, with a 192kbp receptive field Nucleotide Transformer (Dalla-Torre et al., 2023) achieving performance comparable to Enformer (Table 1). The evaluation is performed by training on variant-centered embeddings (see Figure 1a). This approach is analogous to predicting whether a variant can alter expression for any gene, without linking it to the actual gene which is often of biological interest. Many variants with downstream transcriptional effects are located in promoter and enhancer regions. They exert their effect by modulating the binding of transcription factors (Uffelmann et al., 2021), which is a local effect dependant on a short sequence. Notably, many known causal variants from GTEx and the UK Biobank remain significant even when used with contexts of 230bp in MPRA assays

(Siraj et al., 2024). We hypothesize that the causality of a variant, from a variant-centric perspective, can be predicted to a certain extent from the local sequence alone, without the need for a wide receptive field.

We quantified this by using a method similar to that of Tang & Koo (2024), who established naive baselines on top of one-hot encoded DNA sequences. Using the dataset from Section 4.1, we trained a convolutional neural network (CNN) from one-hot encoded sequences with a short local receptive field of just 1500 base pairs (see Section 4.2). The results in Table 1 suggest that the baseline for this task is quite close to the current state-of-the-art achieved by self-supervised models with much larger receptive fields.

Huang et al. (2023) recently evaluated DNA models for their zero-shot capability in predicting gene expression across individuals. However, instead of evaluating at the variant level, they chose to evaluate using a 10-bin window centred at the gene transcription start site (TSS). In a recent pre-print, Linder et al. (2023) created an additional task for evaluating the model's ability to identify the affected gene by the variant from the set of local genes surrounding that variant. Inspired by these studies, we trained an MLP on gene-centered embeddings, using the same ground-truth causal variant dataset (see Section 4.1). We did not necessarily expect *a priori* that gLMs would perform well on this task, as they are not originally trained on expression data. Supervised models like Enformer (Avsec et al., 2021), are

*Table 1.* Results for causal variant prediction using the variant-centric view. *The results for HyenaDNA, Nucleotide Transformer, and Enformer are taken from Kao et al. (2024) and are not recomputed. They are only provided to assist comparison.

| MODEL | BASIC CNN | *HYENADNA | *NUCLEOTIDE TRANSFORMER | *NUCLEOTIDE TRANSFORMER NTK | *ENFORMER |
|---|---|---|---|---|---|
| TRAINED RECEPTIVE FIELD | 1.5KBP | 160KBP | 12KBP | 192KBP | 196KBP |
| AUC | 0.695 | 0.706 | 0.722 | 0.749 | 0.755 |

*Table 2.* Gene-centric results for causal variant prediction across different self-supervised and supervised models on the ground-truth dataset (section 4.1)

| MODEL | HYENADNA | CADUCEUS | ENFORMER |
|---|---|---|---|
| TRAINING RECEPTIVE FIELD | 160KBP | 131KBP | 196KBP |
| INFERENCE RECEPTIVE FIELD | 131KBP | 131KBP | 131KBP |
| GENE EMBEDDING SPAN | 384BP | 384BP | 384BP |
| AUC | 0.670 | 0.703 | 0.764 |

trained to predict gene expression at the TSS.

The results from this approach reveal a performance gap between the current state-of-the-art self-supervised models and supervised models (Table 2), but also indicate a previously unseen, and unexpected, potential capability of self-supervised models. To the best of our knowledge, this analysis, and this gap has not been previously articulated for DNA models.

Due to scaling laws of traditional transformers and computational limitations we were not able to run the Nucleotide Transformer. To provide a second benchmark we used a new recent model called Caduceus (Schiff et al., 2024) which is a bi-directional equivariant version of Mamba (Gu & Dao, 2023).

## 3. Results

### 3.1. Variant-centered evaluation

Utilizing the dataset described in 4.1, we generated 1500 bp DNA windows centered on the variant's location. The DNA sequences for both the reference and alternate alleles were one-hot encoded and input into a CNN architecture as detailed in 4.2. Subsequently, the encoded DNA sequences were concatenated and processed through an MLP. The results of this analysis are presented in Table 1, along with the results from HyenaDNA, Nucleotide Transformer and Enformer from the (Kao et al., 2024).

### 3.2. Gene-centered evaluation

To assess the model's performance in a gene-centric context, the evaluation was focused on a window centred on the gene's TSS. While maintaining a large receptive field at

inference, an embedding narrowly focused around the TSS region was extracted. This embedding was then utilized to train a binary classifier model to predict whether the input variant is causal to the gene. The outcomes of this evaluation are summarized in Table 2.

## 4. Materials and Methods

### 4.1. Ground truth dataset

For this work we used the ground truth dataset from Avsec et al. (2021), which was also used by Kao et al. (2024). Briefly, this dataset contains causal variants from GTEx that have been fine-mapped using the statistical fine-mapping method SuSiE (Wang et al., 2020). Variants with PIP score > 0.9 are labelled as positives. In the download made available by Kao et al. (2024), the affected gene has been removed. We re-added it using the original data from (Avsec et al., 2021). Sometimes this results in multiple gene associations, as a variant-tissue pair can be causal for multiple genes. The Gencode (Frankish et al., 2021) v44 annotation was used to define the genomic location of the TSS for each gene.

### 4.2. CNN for variant-centred evaluation

The encoder for the variant-centred prediction was composed from a series of convolutional blocks. Each convolutional block was set up like in (Avsec et al., 2021), containing a spatial convolution and a pointwise convolution. We used 4 convolutions blocks with the spatial convolutions having kernel sizes 7, 3, 3 and and 5 respectively. The number of filters in each block were 192, 384, 576 and 768 respectively.

The representation for the reference sequence and for the

alternate sequence were concatenated, along with a one-hot encoded tissue vector. These were used as input for an MLP with hidden dimensions 256 and 64.

We trained the model using a binary cross-entropy loss. For training/test splits we adhered to the data splits from Kao et al. (2024), with chromosomes 9 and 10 serving as the held-out set and performed 5-fold cross validation.

### 4.3. Gene-centred evaluation

For this section inference was run on gene-centred windows using 2 self-supervised models: HyenaDNA (Nguyen et al., 2024b), Caduceus (Schiff et al., 2024), and a supervised model Enformer (Avsec et al., 2021). In a similar way to 4.2, we generated embeddings for both the reference and the alternative sequences. We used 131,072 bp window centred on the gene's TSS. As a result we did not consider variants that are further than 65,536bp (131,072/2) away from the TSS. The embedding for the TSS was extracted for a 384bp area around the TSS, which translates to 3 bins for Enformer and 384 bins for HyenaDNA and Caduceus as these generate nucleotide level embeddings.

The logged absolute difference of the reference and alternate embeddings was used as input for an MLP. The MLP had 2 hidden layers of sizes 256 and 64, and was trained with a binary cross entropy loss. As in 4.2 we used 5-fold cross validation using chromosome 9 and 10 as held out sets.

For this task we did not benchmark the Nucleotide Transformer models as these used a traditional transformer architectures and for large receptive fields the model does not scale within A100 GPUs.

## 5. Discussion

The variant-centric approach has traditionally been the standard for evaluating DNA models on causal variant prediction. In this study, we demonstrated that results close to the state-of-the-art can be achieved using basic convolutional neural networks. We proposed a novel gene-centric evaluation method, which revealed previously unknown strengths of genomic language models (gLMs), while also highlighting a performance gap compared to current state-of-the-art supervised models.

Our evaluations have certain limitations. Using only the embeddings for these models might not fully capture the power of gLMs, as they were not specifically trained to predict gene expression. Future research should explore fine-tuning these foundational models specifically for gene expression prediction tasks. Additionally, our analysis focused on a narrow window around the TSS. Supervised models are likely better predictors at the TSS because of the datasets they were originally trained on. For gLMs, there may be

better regions to test or more effective ways to prompt these models to reveal long-range downstream effects. Integrating both supervised and self-supervised methods in end-to-end training could further enhance model performance.

The benchmark itself may have limitations. We found that the tissue dimension can predict whether genes are causal or non-causal to a small extent, which can influence results on benchmarks. Ensuring that the negative examples are sufficiently challenging, focusing on those within the same credible set and at similar distances from the TSS, is crucial for accurate evaluation.

Our findings outline the need for further research to assess model performance and develop robust benchmarks. This will ultimately enhance our ability to link genetic variation to functional outcomes. We anticipate that the proposed gene-centric benchmark will assist the machine learning community in achieving this objective and advancing the field of genomic research.

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
