# OpenReview forum: "Gene-centric evaluation of causal variant prediction for DNA models"
_ICML.cc/2024/Workshop/ML4LMS — ML4LMS Poster_

### Official Review · Reviewer_y4p7 · 2024-06-05
**Gene-centric evaluation of causal variant prediction for DNA models**

**Rating:** 6
**Confidence:** 4

**Review:**

The methodology is implemented well and well presented. However, the current work has an incremental advancement and comparing other existing models, it performs at the same rate.

---

### Official Review · Reviewer_SZjM · 2024-06-11
**important problem**

**Rating:** 7
**Confidence:** 4

**Review:**

Important problem. Hope to provide comparisons with different approaches.

---

### Official Review · Reviewer_Q2TZ · 2024-06-12
**Benchmarking DNA foundation models' ability to identify genes affected by non-coding variants**

**Rating:** 5
**Confidence:** 3

**Review:**

This paper compares the performance of self-supervised DNA foundation models and supervised DNA->functional property prediction models on the task of identifying which gene is causally affected by a given non-coding variant. Although this is an interesting comparison, the task itself is not new (it is adapted, as stated by the authors, from an evaluation performed in the Borzoi paper), and the results are broadly in line with what would be expected from previously reported results on the related task of predicting whether a non-coding variant has a significant effect on some gene, limiting the significance and originality of the work.

Clarifications:
* Is the number of negatives the same for each variant? If not, how does this affect results?
* Are the embeddings pooled over the bins before being passed through the MLP?